# HOW CAN LLM GUIDE RL? A VALUE-BASED AP-PROACH

## ABSTRACT

Reinforcement learning (RL) has become the de facto standard practice for sequential decision-making problems by improving future acting policies with feedback. However, RL algorithms may require extensive trial-and-error interactions to collect useful feedback for improvement. On the other hand, recent developments in large language models (LLMs) have showcased impressive capabilities in language understanding and generation, yet they fall short in exploration and self-improvement capabilities for planning tasks, lacking the ability to autonomously refine their responses based on feedback. Therefore, in this paper, we study how the policy prior provided by the LLM can enhance the sample efficiency of RL algorithms. Specifically, we develop an algorithm named `LINVIT` that incorporates LLM guidance as a regularization factor in value-based RL, leading to significant reductions in the amount of data needed for learning, particularly when the difference between the ideal policy and the LLM-informed policy is small, which suggests that the initial policy is close to optimal, reducing the need for further exploration. Additionally, we present a practical algorithm `SLINVIT` that simplifies the construction of the value function and employs sub-goals to reduce the search complexity. Our experiments across three interactive environments—ALFWorld, InterCode, and BlocksWorld—demonstrate that the proposed method achieves state-of-the-art success rates and also surpasses previous RL and LLM approaches in terms of sample efficiency.

## 1 INTRODUCTION

Trained on the web-scale corpora, Large Language Models (LLMs) have exhibited emergent capabilities and seen tremendous success across various fields, such as code development (Chen et al., 2021; Roziere et al., 2023; Li et al., 2023) and theorem proving (Yang et al., 2023b; Romera-Paredes et al., 2023). The recent advances in robotics (Huang et al., 2023b; Liang et al., 2023) and games (Wang et al., 2023a; Yuan et al., 2023; Wang et al., 2023c; Liu et al., 2023b) further highlight the potential of LLMs to build effective agents in well-designed interactive environments.

However, the reasoning and planning abilities of LLMs, which are important for intelligent agents, have been found to be inconsistent and often unreliable (Valmeekam et al., 2023b; Mahowald et al., 2023; Huang et al., 2023a; Pallagani et al., 2023). Besides, agents powered by LLMs tend to have limited abilities to explore different strategies, frequently defaulting to repeating established policies. This limitation becomes particularly pronounced in complex decision-making scenarios that LLMs are not specifically attuned to, resulting in significant difficulties in refining their strategies based on environmental feedback by reasoning the environment feedback based solely on its inherent capabilities (Valmeekam et al., 2023a; Shinn et al., 2023; Ivanova, 2023; Zhang et al., 2024). On the contrary, Reinforcement Learning (RL) is a well-studied methodology for improving future acting policies with feedback. Unfortunately, improving from scratch without the guidance of prior knowledge, such as common sense, requires the RL agents to take a huge amount of random interactions to collect useful feedback, leading to poor sample efficiency and even failure in sparse-reward environments.

Hence, in this paper, we aim to tackle these issues and answer the following question:

*Can we improve the sample efficiency of Reinforcement Learning with Large Language Models?*

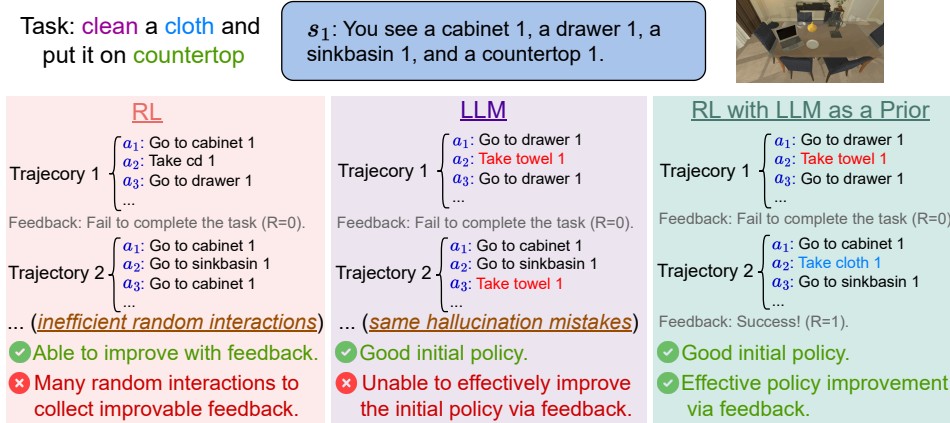

Figure 1: Illustration of the differences and the respective advantages, disadvantages of RL and LLM agents in an instance of the ALFWorld decision-making task. We propose an RL framework leveraging the LLM as a policy prior that gets the best of both worlds.

Our primary objective is to develop an algorithm that is both theoretically robust and empirically effective, utilizing LLMs to enhance sample efficiency. Our pivotal insight is the utilization of LLMs to define a regularizer, as opposed to directly employing them in decision-making. Leveraging the properties of regularized-MDPs, we find that sample complexity can be significantly reduced when the LLM-provided policy closely aligns with the optimal policy. Moreover, our approach retains the capability to identify the optimal policy even in scenarios where the LLM policy falls short. An illustration comparing the standard RL algorithms, LLM agents, and the RL framework with LLM as a prior is shown in Figure 1. To demonstrate this concept, we introduce an algorithm named *Language-INtegrated Value Iteration* (LINVIT), which shows a marked improvement in sample complexity, particularly when the Kullback-Leibler (KL) divergence between the optimal policy and the LLM policy is minimal.

We further present a practical algorithm called SLINVIT and empirically validate it in various benchmarks, including ALFWorld (Shridhar et al., 2020), the interactive coding environment InterCode (Yang et al., 2023a), and the planning benchmark BlocksWorld (Valmeekam et al., 2023b). Experimental results show that the proposed algorithm outperforms previous RL and LLM algorithms by a large margin, achieving higher success rates with fewer numbers of samples.

## 2 BACKGROUND

**Reinforcement Learning.** Consider the problem of learning to optimize a finite $H$-horizon Markov Decision Process (MDP) over repeated episodes of interaction. We denote by $\mathcal{S}$ and $\mathcal{A}$ the state and action space, respectively. When taking an action $a \in \mathcal{A}$ at a state $s \in \mathcal{S}$ at timestep $h$, the agent receives a reward $r_h(s, a)$ and the MDP transits to a new state $s'$ according to $s' \sim P_h^*(\cdot \mid s, a)$.

We aim to find a policy $\pi$ that maps a state to an action distribution to maximize the expected cumulative reward. We denote by $Q_h^* : \mathcal{S} \times \mathcal{A} \to \mathbb{R}$ and $V_h^* : \mathcal{S} \to \mathbb{R}$ the state-action value function and the state value function associated with $\pi$, respectively, which are defined as follows,

$$Q_h^*(s, a) = r_h(s, a) + \sum_{s'} P_h^*(s' \mid s, a) V_{h+1}^t(s'), \qquad V_h^*(s) = \sum_a \pi(a \mid s) Q_h^*(s, a),$$

where $s \in \mathcal{S}, a \in \mathcal{A}$. The objective of the decision-making problem is to maximize the state value at the initial timestep $V_1^*(s_1)$, where $s_1 \in \mathcal{S}$ is the initial state.

**Decision-Making with LLMs.** To solve decision-making problems, one can prompt the LLM agent to generate action responses $\pi_h^{\text{LLM}}(a \mid s)$ based on its state $s$, which consists of the observation-action history up to the current timestep $h$ in partially observable MDPs, or contains an additional reasoning step (Yao et al., 2022) in a chain-of-thought (Wei et al., 2022) manner. Unfortunately, without the domain knowledge of a specific task or environment, the LLM policy is

hard to be optimal, especially when the planning problems have long horizons and the LLM lacks the necessary reasoning abilities. In this work, we explore another manner of leveraging the LLM policy as the priorin RL.

# 3 USING LANGUAGE MODEL AS A POLICY PRIOR

In this section, we present an algorithm leveraging LLM to enhance sample efficiency. We begin by discussing the algorithm's motivation, followed by a detailed explanation of its procedure.

**Motivation.** A simplistic approach to using Large Language Models (LLMs) in decision-making is directly applying the LLM-generated policy to target tasks. However, pretrained and fine-tuned on the static datasets, LLMs are not inherently attuned to the specific interactive environments of concern and are unable to modify their policies based on environmental feedback. Consequently, effectively leveraging LLM information for decision-making remains an unresolved challenge. Inspired by the property of entropy-regularized MDP (Neu et al., 2017), our approach employs the large language model as a supplemental regularizer within the original algorithm, rather than using it as the primary decision-making tool. This methodology significantly improves sample efficiency when the LLM's policy is closely aligned with the optimal policy. Moreover, we can still identify the optimal policy for the original MDP even if the LLM's policy is suboptimal.

---

**Algorithm 1** Language-INtegrated Value Iteration(`LINVIT`)

---

**Input:** Target precision $\epsilon$, target probability $\delta$, bonus function $b^0$, $b^{0,\mathrm{KL}}$.
1: **for** $t = 0, \dots, T$ **do**
2:     Construct the model estimator $P_h^t$ and $u_h^t$ as (3.1).
3:     Compute the optimistic and pessimistic value $\overline{V}_h^t$ and $\underline{V}_h^t$ as (3.2) and (3.3).
4:     Compute $\pi^t$ as (3.4).
5:     **for** $h = 1, \dots, H$ **do**
6:         Sample $a_h^t \sim \pi_h^t(\cdot|s_h^t)$, and observe $s_{h+1}^t$ from the environment.
7:     **end for**
8: **end for**
9: Return $\widehat{\pi}$, which the uniform mixture of $\{\bar{\pi}^t\}_{t=1}^T$.

---

With the above motivation, we propose a novel algorithm *Language-INtegrated Value Iteration* (`LINVIT`), which is an iterative algorithm that outputs a policy after $T$ iterations. In each iteration $t \in [T]$, we first utilize the gathered data to estimate the transition model and calculate the uncertainty associated with our estimation. This estimator is employed to formulate both the optimistic and pessimistic regularized value functions. The final step of each iteration involves leveraging these value functions to develop an exploration policy, which is then used to acquire additional data from the environment. We summarize our algorithm in Algorithm 1. In the following part, we elaborate each of the above steps in detail.

**Model and Uncertainty Estimation.** We estimate the transition model as follows. Let $n_h^t(s, a)$ denote the number of times the state-action pair $(s, a)$ has been visited at step $h$ during the first $t$ episodes, and let $n_h^t(s, a, s')$ denotes the number of times the state-action-next-state triplet $(s, a, s')$ at the same step and episode count. Our dynamics estimator $P_h^t(s'|s, a)$ is defined as $P_h^t(s'|s, a) = n_h^t(s, a, s')/n_h^t(s, a)$ if $n_h^t(s, a) > 0$ and $P_h^t(s'|s, a) \triangleq 1/S$ for all $s' \in \mathcal{S}$ else. We then define the uncertainty quantifier $u_h^t$ by

$$u_h^t(s, a) \triangleq \max\left\{2H, \sqrt{\frac{\log(4HTS^2A/\delta)}{n_h^t(s, a)}}\right\}. \tag{3.1}$$

Intuitively, the uncertainty quantifier $u_h^t$ is inversely related to the frequency of visits to a state-action pair; the less frequently a state-action pair is visited, the greater the value of $u_h^t$. This relationship means that $u_h^t$ effectively measures our uncertainty regarding each state-action pair, capturing the uncertainty of our estimation.

**Regularized Value Functions.** After estimating the transition model and the uncertainty, we compute the optimistic and the pessimistic regularized value function by

$$\overline{Q}_h^t(s,a) = \text{clip}\Big\{r_h(s,a) + \sum_{s'} P_h^t(s' \mid s,a)\overline{V}_{h+1}^t(s') + u_h^t(s,a)\Big\},$$

$$\overline{V}_h^t(s) = \max_{\pi \in \Delta_A}\Big\{\sum_a \pi(a|s)\overline{Q}_h^t(s) - \lambda \text{KL}\big(\pi(\cdot \mid s)\|\pi_h^{\text{LLM}}(\cdot \mid s)\big)\Big\}, \tag{3.2}$$

with $\overline{V}_{H+1}^t = \underline{V}_{H+1}^t = 0$ by convention, and $\text{clip}(x) = \min\Big\{\max\{x,0\},H\Big\}$. The definition of $\overline{Q}$ and $\overline{V}$ comprise three components: the expected reward, the uncertainty estimator, and the regularization defined via $\pi^{\text{LLM}}$. It can be viewed as an optimistic estimation of the regularized value function. We similarly define $\underline{Q}$ and $\underline{V}$ as

$$\underline{Q}_h^t(s,a) = \text{clip}\big(r_h(s,a) + \sum_{s'} P_h^t(s' \mid s,a)\overline{V}_{h+1}^t(s') - u_h^t(s,a)\big),$$

$$\underline{V}_h^t(s) = \max_{\pi \in \Delta_A}\Big\{\sum_a \pi(a|s)\underline{Q}_h^t(s,a) - \lambda \text{KL}\big(\pi(\cdot \mid s)\|\pi_h^{\text{LLM}}(\cdot \mid s)\big)\Big\}. \tag{3.3}$$

Similar to $\overline{Q}$ and $\overline{V}$, $\underline{Q}$ and $\underline{V}$ can be regarded as pessimistic estimations of the regularized value function. The primary distinction between $\underline{Q}$ and $\overline{Q}$ lies in the sign of the uncertainty estimator.

**Sampling Policy.** We explore the environment and collect data with $\pi^{t+1} = \{\pi_h^{t+1}\}_{h=1}^H$, which is defined as

$$\pi_h^t(\cdot \mid s) = \frac{1}{H} \cdot \mathbb{1}\{a = \text{argmax}\, \overline{Q}_h^t(s,a) - \underline{Q}_h^t(s,a)\} + \frac{H-1}{H} \cdot \bar{\pi}_h^t(\cdot \mid s), \tag{3.4}$$

$$\text{where } \bar{\pi}_h^t(\cdot \mid s) = \underset{\pi \in \Delta_A}{\text{argmax}}\Big\{\sum_a \pi(a \mid s)\overline{Q}_h^t(s) - \lambda \text{KL}\big(\pi(\cdot \mid s)\|\pi_h^{\text{LLM}}(\cdot \mid s)\big)\Big\}.$$

Intuitively, the difference $\overline{Q}_h^t(s,a) - \underline{Q}_h^t(s,a)$ captures the uncertainty of the estimation of the regularized value function. As a result, the policy $\pi_h^t$ is designed to act predominantly as a greedy policy concerning the optimistic regularized value function, doing so with a probability of $1 - 1/H$. Conversely, with a probability of $1/H$, it opts for the action associated with the greatest uncertainty. This approach ensures a balance between exploiting known rewards and exploring actions with higher uncertainty to refine the value function estimation.

# 4 RELATED WORK

**Reinforcement Learning with Language.** Language offers a particularly effective medium for tackling decision-making challenges due to its succinct and structured format. This quality has made it a valuable tool for numerous reinforcement learning (RL) algorithms, enabling them to learn from high-level specifications of goals (Jiang et al., 2019; Lynch & Sermanet, 2020b; Hejna et al., 2023) or to benefit from the step-by-step instructions provided by large language models (LLMs) (Ahn et al., 2022; Huang et al., 2022). Additionally, research has ventured into harnessing more expansive language applications to model the dynamics and reward mechanisms of environments, employing planning algorithms to guide decision-making processes (Bialystok, 1978; Liu et al., 2023b). Different from these approaches, our work introduces the novel concept of applying LLMs as regularizing agents within value-based RL frameworks.

**Decision-Making with Language Models.** The strong capabilities language models exhibit have opened a new avenue for LLM agents to interact with the real world autonomously for decision-making tasks. Inspired by classical planning literature (Bonet & Geffner, 2001; Hoffmann & Nebel, 2001; Chitnis et al., 2016; Gehring et al., 2022) that uses heuristic functions as dense reward generators to perform informed search, recent works (Lin et al., 2023; Hao et al., 2023) proposed to use the LLM as the heuristic function. The remarkable programming abilities exhibited by the LLM have also enabled converting natural language instructions into planning languages and then adopting the classical planner (Liu et al., 2023a; Liang et al., 2023; Silver et al., 2023; Xie et al., 2023), which,

however, are constrained in narrowed domains and predefined environments. Moreover, a recent line of work (Yao et al., 2023a;b; Sel et al., 2023; Zhang et al., 2023) has developed various prompting schemes to enhance LLM reasoning, though these approaches generally do not integrate feedback from the environment into the decision-making process.

A large body of previous works focused on prompt engineering by providing the LLM with additional contextual information and templates to complete the task. Among them, the ReAct (Yao et al., 2022) agent generates both reasoning traces and task-specific actions in an interleaved manner, Plan-and-Solve (Wang et al., 2023b) improves the Chain-of-Thought (Wei et al., 2022) prompt to devise a fixed high-level plan before taking actions in the environment, and Huang et al. (2022) prompt the LLM to extract temporally extended plans in a zero-shot manner. Besides, other works directly train the model on embodied decision-making data (Suglia et al., 2021; Sharma et al., 2021; Mezghani et al., 2023; Driess et al., 2023) or multi-modal data (Lu et al., 2019; Li et al., 2019; Radford et al., 2021; Zellers et al., 2021b) by learning additional downstream networks on top of the pre-trained LLM (Lynch & Sermanet, 2020a; Akakzia et al., 2020; Zellers et al., 2021a) or finetuning in the environment (Reid et al., 2022; Li et al., 2022; Chen et al., 2023). Similar to our work, Ahn et al. (2022); Hu & Sadigh (2023); Lin et al. (2023); Hao et al. (2023) also use value functions to ground the LLM agent, but the sub-problem horizon is set to 1 and the executed actions are one-step greedy without backtracking, which still suffers from the curse of the long horizon. Works building on the self-reflection abilities of LLMs (Shinn et al., 2023; Sun et al., 2023; Ma et al., 2023) also demonstrate limitations in refining initial strategies based on feedback, as shown in our experiments.

Moreover, there are various approaches that fine-tune the LLM policies to maximize the cumulative reward using RL. Examples include GLAM (Carta et al., 2023) and ETPO (Wen et al., 2024), which employs a token-level entropy-augmented RL method. Additionally, ILQL (Snell et al., 2022) uses the LLM as a perturbation for Q-values in offline RL. In contrast, our method integrates value iterations into the LLM for effective policy improvement via feedback without fine-tuning.

## 5 EXPERIMENTS

In this section, we conduct empirical studies in several text-based benchmarks, including the embodied environment ALFWorld (Shridhar et al., 2020), the interactive coding environment InterCode (Yang et al., 2023a), and the standard planning benchmark BlocksWorld (Valmeekam et al., 2023b). Across these three benchmarks, we measure the algorithm's effectiveness by its success rate, which we define as the ratio of the number of task instances the algorithm successfully completes. More precisely, each task instance is defined by a target state $s_g \in \mathcal{S}$, and a task is deemed successfully completed if the algorithm reaches this target state $s_H = s_g$ at the end.

We will first delve into a detailed discussion of our implementation approach. Following this, we will present and analyze the results of our experiments.

---

**Algorithm 2** Simplified Language-INtegrated Value Iteration (SLINVIT)

---

**Input:** Sub-problem horizon $N$, BFS breadth $k$.
1: Construct the value estimator $\widehat{V}$ (rule-based or Monte-Carlo)
2: **for** $i = 1, \ldots, H/N$ **do**
3:     Solve (5.1) with breadth-$k$ BFS
4:     Execute the resulting $a_{(i-1)N+1:iN}$
5: **end for**

---

### 5.1 IMPLEMENTATION DETAILS

In our experiments, we introduce two primary simplifications to the LINVIT algorithm to enhance its efficiency and practicality. These modifications lead to a streamlined variant we denote as SLINVIT, detailed in Algorithm 2. Below, we elaborate on each simplification:

**Construction of Value Function and Exploration Policy.** In Algorithm 1, we use a bonus in the construction of the value function and combine the uniform policy with the optimal policy in the regularized MDP in the exploration policy. This approach, while sample-efficient, introduces significant computational complexity. To optimize computation in our experiments, we simplify these

processes. Specifically, we directly combine the original value estimator $\widehat{V}$ with the log probability of the LLM policy $\mathbb{P}_{\text{LLM}}$, which is the key component in (3.2), to construct the value estimator in the experiment. Furthermore, we adopt a greedy policy with respect to this adjusted, regularized value estimator. This simplification enables a simpler computation by focusing on the core elements that drive the decision-making process.

**Using Sub-Goals to Reduce Searching Complexity.** Since the complexity of directly searching for the maximizor of the regularized value function is exponential in the horizon $H$, we leverage sub-goal states to reduce the searching complexity. In the experiment, our algorithm works by decomposing the $H$-horizon planning problem into $H/N$[1] sub-problems, each of which has a sub-goal and is of horizon $N$. More specifically, for each sub-problem $i \in [1, H/N]$, the corresponding sub-goal $s_{iN+1}$ is determined by solving

$$Q^{\text{LLM}}(s_{(i-1)N}, a_{(i-1)N+1:iN}) := \widehat{V}(s_{iN+1}) + \sum_{h=(i-1)N+1}^{iN} \lambda \pi_h^{\text{LLM}}(a_h|s_h),$$

$$a_{(i-1)N+1:iN} := \underset{a_{(i-1)N+1:iN}}{\operatorname{argmax}} \; Q^{\text{LLM}}(s_{(i-1)N}, a_{(i-1)N+1:iN}). \tag{5.1}$$

where $\widehat{V}$ is the estimator of the true value $V^\pi$ and $\lambda \geq 0$ is a hyperparameter for the regularization. Compared with the regularized value function $\overline{V}$ in Section 3, we remove the logarithm term before $\mathbb{P}_{\text{LLM}}$ for stability, such that it has a similar scale with $r \in [0, 1]$. Here, $\widehat{V}$ can take various forms depending on both the true policy value it estimates and its own approximators, which we will discuss in more detail in Section 5.2.

In practice, we implement a breadth-$k$ Breadth First Search (BFS) to approximate the actions in (5.1). Specifically, the following procedure is repeated $N$ times: making $k$ copies of the agent that execute the top-$k$ outputs of the LLM by querying it "What is the potential next-step action?". This will generate $k^N$ lookahead action sequences and the one with the highest $Q^{\text{LLM}}$ is selected as $a_{(i-1)N+1:iN}$ and executed in the environment.

## 5.2 INSTANTIATIONS OF VALUE ESTIMATOR

In this section, we describe two instantiations of the value estimator $\widehat{V}$ in (5.1), named rule-based and Monte-Carlo value estimators. An illustration is provided in Figure 2.

**Rule-Based Value Estimation.** The rule-based value estimator is designed for scenarios where achieving the goal $s_g$ requires fulfilling multiple preconditions, such as in ALFWorld and BlocksWorld. It outputs the ratio of preconditions currently met by the state $s$. To achieve this, we prompt the LLM with "Estimate the value of the task by generating Python functions" as well as the task description ahead of evaluation. To avoid uncontrollable mistakes of the LLM, its response undergoes a one-time human review. This step is necessary only once because, although $s_g$ is changing during evaluation (e.g., "put a cup on table" and "put a pen in drawer"), the nature of the task and the structure of the preconditions (e.g., "pick A" and "place A on B" for any "put" task) do not vary. We provide how we implement this in Appendix C.

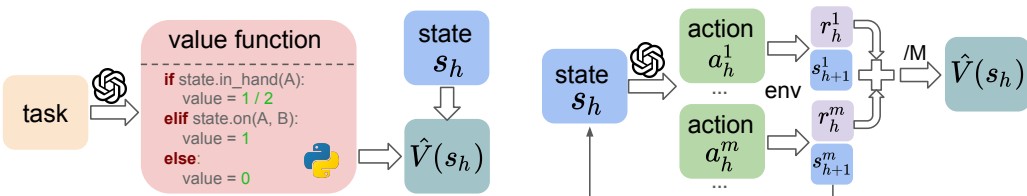

(a) Rule-based value estimator.      (b) Monte-Carlo value estimator.

Figure 2: Illustration of the proposed two instantiations of the value estimator.

For tasks where the goal's preconditions are hard to determine, we propose the more general Monte-Carlo estimator.

---

[1]For notation convenience, we assume w.l.o.g. that $h$ can be divided by $N$.

**Monte-Carlo Value Estimation.** At the state $s_h$ to be evaluated, by sampling actions from the LLM policy $\pi^{\text{LLM}}(\cdot \mid s_h)$ until the planning horizon $H$ is reached, we obtain a partial trajectory. The Monte-Carlo value estimation is then given by averaging the cumulative reward received in $M$ such rollouts to approximate the value of the LLM policy, i.e.,

$$\widehat{V} = \frac{1}{M} \sum_{m=1}^{M/(H-h)} \sum_{n=h}^{H} r_h(s_n^m, a_n^m), \qquad \text{where } a_n^m \sim \pi_n^{\text{LLM}}(\cdot \mid s_n^m) \text{ and } s_{n+1}^m \sim P_n^*(\cdot \mid s_n^m, a_n^m).$$

## 5.3 ALFWORLD

ALFWorld (Shridhar et al., 2020) is an interactive text-based environment with aligned embodied simulators. The benchmark encompasses 134 virtual household task instances with predefined and fixed goals, each of which can be categorized into one of the six task types as shown in Table 1.

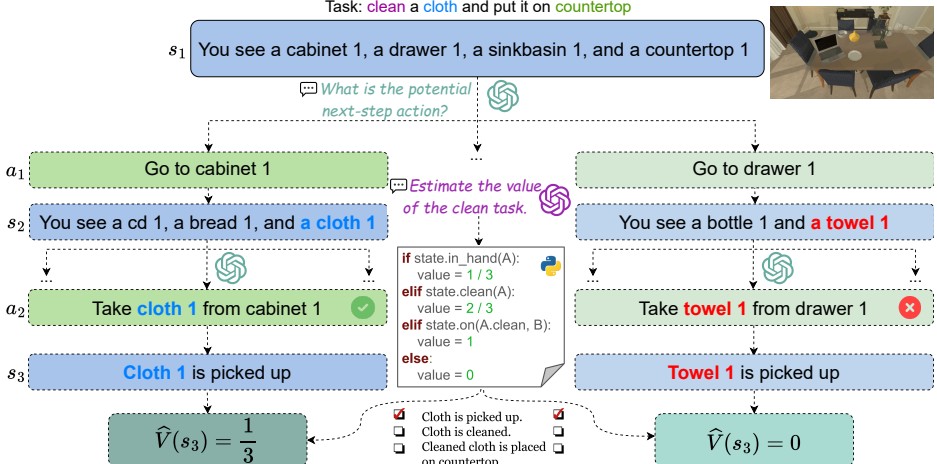

Figure 3: Demonstration of the `SLINVIT` algorithm in the ALFWorld environment when $N = 2$ and the tree breadth of BFS is set to $k = 3$. The task is to "`clean a cloth and put it on countertop`". The hallucination that LLM faces, i.e., the towel should be taken (instead of cloth), is addressed by the inherent exploration mechanism in our RL framework.

We provide a visualization of how `SLINVIT` works in the ALFWorld environment in Figure 3. Specifically, for each type of the six tasks, we use the rule-based value estimator to generate Python code that determines the preconditions of the task goal and the current state. The code outputs the portion of the satisfied preconditions as the estimated value. We set $N = 2$ in our implementation for the ALFWorld benchmark.

| | Pick | Clean | Heat | Cool | Examine | PickTwo | Total |
|---|---|---|---|---|---|---|---|
| BUTLER | 46.00 | 39.00 | 74.00 | **100.00** | 22.00 | 24.00 | 37.00 |
| ReAct | 66.67 | 41.94 | 91.03 | 80.95 | 55.56 | 35.29 | 61.94 |
| AdaPlanner | **100.00** | 96.77 | **95.65** | **100.00** | **100.00** | 47.06 | 91.79 |
| Reflexion | **100.00** | 90.32 | 82.61 | 90.48 | **100.00** | 94.12 | 92.54 |
| SLINVIT | **100.00** | **100.00** | 91.30 | 90.48 | **100.00** | **100.00** | **97.01** |

Table 1: Success rate (%) comparison of `SLINVIT` and baselines, including LLM agents and RL algorithms, in the ALFWorld environment.

We compare the success rate of `SLINVIT` and baselines in the ALFWorld environment in Table 1. We use GPT-3 (`text-davinci-003`) in our implementation. All the LLM agent baselines, including `ReAct` (Yao et al., 2022), `AdaPlanner` (Sun et al., 2023), and `Reflexion` (Shinn et al., 2023), use GPT-3 that is the same as ours, while `BUTLER` (Shridhar et al., 2020) is a RL-style imitation learning

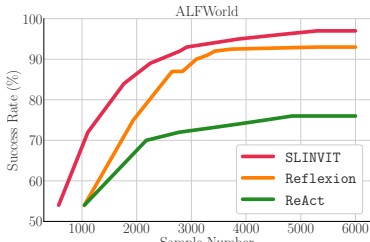

Figure 4: Success rates with different numbers of samples.

|  | SQL Sample Number | | | Bash Sample Number | | |
|---|---|---|---|---|---|---|
|  | 10 | 20 | 30 | 10 | 20 | 30 |
| TryAgain | 48.45 | 50.97 | 50.97 | 34.67 | 40.20 | 50.25 |
| ReAct | 52.61 | 52.71 | 53.67 | 20.50 | 21.10 | 21.61 |
| SLINVIT | **52.80** | **58.51** | **64.02** | **46.23** | **50.25** | **54.27** |

Table 2: Success rate (%) under different maximum numbers of samples per episode in the InterCode-SQL and InterCode-Bash environments.

algorithm. Notably, the inferior performance of BUTLER indicates that RL algorithms may have difficulties understanding the task and generalize beyond. On the contrary, SLINVIT achieves the highest success rate in most categories of the task types and outperforms the baselines in terms of the overall success rate.

We also report the changes in success rate when the numbers of samples are different. The results are shown in Figure 4. The data points of SLINVIT are obtained by changing the tree breadth when performing BFS. Specifically, we set the tree breadth to $k = 2, 3, \cdots, 10$ and report the overall success rate corresponding to different numbers of samples in all the 134 tasks. For the Reflexion and ReAct baselines, the points in the plot are the results at the end of each trial. The sample number is calculated as the number of samples taken in the successful tasks in the current trial, plus all the samples in the previous trials. We observe that the proposed algorithm is able to achieve a higher success rate with fewer numbers of samples compared to methods that incorporate the environment feedback summary into the LLM as additional context.

## 5.4 INTERCODE

InterCode (Yang et al., 2023a) is an interactive coding benchmark with code or command as actions and the environment feedback after executing an action as observations. It provides two benchmarks to evaluate the planning abilities of the large language models, namely InterCode-SQL and InterCode-Bash which use SQL and Bash commands as action spaces, respectively. For each benchmark, there are hundreds of tasks with predefined goals, such as "*Find the name of airports which do not have any flight in and out.*" and "*Find all text files in the testbed directory and subdirectories and concatenate them into a single file.*".

Unlike the ALFWorld environment where the task goals can be described by preconditions, there is no straightforward way to directly measure the value of the current state with explicit and simple rules in the InterCode environment. Therefore, we implement SLINVIT using the Monte-Carlo value estimator. Besides, we use the original dense reward as proposed in (Yang et al., 2023a). We set the sub-problem horizon $N = 1$ and the Monte-Carlo sampling number $M = 1$. For our method and all the baselines, we use GPT-3.5 (gpt-3.5-turbo).

|  | InterCode-SQL Hardness | | | | |
|---|---|---|---|---|---|
|  | Easy | Medium | Hard | Extra | Total |
| $M = 1$ | **90.73** | 71.08 | 60.34 | 50.00 | 70.60 |
| $M = 2$ | 88.31 | **77.13** | **65.52** | **52.42** | **73.89** |

Table 3: Ablation study on SLINVIT with different $M$.

The success rates in the InterCode-SQL and InterCode-Bash environments are reported in Table 4. The baselines we compare include ReAct (Yao et al., 2022), Plan & Solve (Wang et al., 2023b), and Try Again (Yang et al., 2023a), which is a vanilla LLM-based planning algorithm. We observe that SLINVIT achieves the highest success rate in all the hardness modes of the InterCode-SQL benchmark, all the file systems of the InterCode-Bash benchmark, and has the best overall performance. In Table 2, we investigate the sample efficiency of the proposed algorithm in the InterCode environment. Specifically, we set the maximum number of samples in each episode to be 10, 20, and 30

| | InterCode-SQL Hardness | | | | | InterCode-Bash File System | | | | |
| --- | --- | --- | --- | --- | --- | --- | --- | --- | --- | --- |
| | Easy | Med. | Hard | Extra | Total | Sys 1 | Sys 2 | Sys 3 | Sys 4 | Total |
| TryAgain | 75.81 | 48.65 | 49.43 | 21.69 | 50.97 | 45.00 | 49.06 | 45.00 | 48.15 | 46.50 |
| Plan & Solve | 77.42 | 49.78 | 32.18 | 22.89 | 49.13 | 0.00 | 45.28 | 43.33 | 22.22 | 28.00 |
| ReAct | 80.24 | 65.47 | 47.13 | 20.48 | 58.70 | 21.67 | 5.66 | 30.00 | 25.93 | 20.50 |
| SLINVIT | **90.73** | **71.08** | **60.34** | **50.00** | **70.60** | **55.00** | **60.38** | **64.41** | **66.67** | **60.80** |

Table 4: Success rate (%) comparison in the InterCode-SQL and InterCode-Bash environment.

and report the corresponding success rates. For SLINVIT, both the samples taken to maximize (5.1) and the samples for Monte-Carlo value estimation are counted. The results indicate that SLINVIT consistently outperforms the baselines with the same sample size and is thus more sample-efficient.

**Ablation.** We conduct an ablation study on the number of rollouts $M$ and the results are shown in Table 3. We observe that a more accurate value estimation corresponding to a larger $M$ leads to higher success rates for harder problems. Therefore, a trade-off can be taken between the sample number and the performance.

## 5.5 BLOCKSWORLD

BlocksWorld (Valmeekam et al., 2023b; Liu et al., 2023a) is another planning benchmark that contains various tasks to arrange blocks in specific configurations. The state is the current configuration of the blocks and the action is an instruction that moves blocks. Specifically, an action is composed of one of the four verbs (STACK, UNSTACK, PUT, and PICKUP) and the operated block. Similar to the implementation in ALFWorld, we also adopt the rule-based value estimator. Specifically, the goal state of each task is defined as the combination of several block arrangements (e.g., "block 1 is on top of block 2"). With the current state as input, the value estimator then returns the proportion of the satisfied arrangements.

Following RAP (Hao et al., 2023), we group the task instances in Valmeekam et al. (2023b) by the minimum number of actions required, resulting in 57 cases that are solvable within 4 steps, and 114 cases that are solvable within 6 steps. We evaluate our method and the RAP baseline by comparing the success rates under different numbers of samples. We evaluate the Vicuna-13b(v1.3) model and the results are shown in Figure 5. Our method consistently outperforms RAP and achieves a higher success rate with fewer samples.

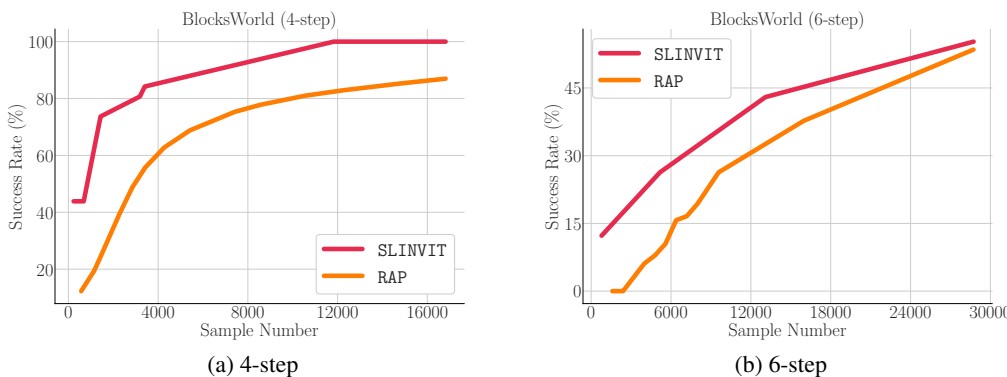

(a) 4-step      (b) 6-step

Figure 5: Success rate (%) of SLINVIT and baselines on the 4-step and 6-step BlocksWorld tasks.

## 6 THEORY

In this section, we present the analysis of LINVIT. To begin, we first define the Kullback–Leibler (KL) divergence between two policies as follows:

**Definition 6.1** (KL-divergence between two policy). For two policies $\pi_1 = \{\pi_1^h\}_{h=1}^H$ and $\pi_2 = \{\pi_2^h\}_{h=1}^H$, we define

$$\mathrm{KL}(\pi^1\|\pi^2) = \sum_{h=1}^H \mathbb{E}_{\pi^1}\Big[\mathrm{KL}\big(\pi_h^1(\cdot|s_h)\|\pi_h^2(\cdot|s_h)\big)\Big].$$

Definition 6.1 provides a quantitative measure of the similarity between two policies. More specifically, the divergence is small when two policies are similar. Building on this foundational understanding, we present the following theorem.

**Theorem 6.2.** We assume that $\mathrm{KL}(\pi^*\|\pi^{\mathrm{LLM}}) \le \epsilon_{\mathrm{LLM}}$, and set the tuning parameter

$$\lambda = \epsilon/(2\epsilon_{\mathrm{LLM}}), \quad T = CH^6SA^4\log^2(HSA/\delta)\epsilon_{\mathrm{LLM}}/\epsilon^2$$

for some absolute constant $C$. We then have $V_1^*(s_1) - V_1^{\widehat{\pi}}(s_1) \le \epsilon$ with probability as least $1 - \delta$.

*Proof.* See Appendix §A for a detailed proof. □

Theorem 6.2 demonstrates that the number of samples required to achieve $\epsilon$-optimality is proportional to the KL divergence, $\mathrm{KL}(\pi^*\|\pi^{\mathrm{LLM}})$, given that $\lambda$ is suitably chosen. This relationship implies a reduced sample necessity when $\pi^{\mathrm{LLM}}$ closely aligns with the optimal policy $\pi^*$. The intuitive rationale behind this is that the demand for exploration diminishes when an initial policy is nearly optimal. Consequently, this theorem underscores the efficacy of our algorithm in capitalizing on the policy information provided by the Large Language Model (LLM), thereby validating its practical utility in decision-making scenarios. Theorem 6.2 further demonstrates that Algorithm 1 is capable of achieving $\epsilon$-optimality, even in cases where $\epsilon \le \epsilon_{\mathrm{LLM}}$, contingent upon the collection of a sufficient number of samples. This finding underscores the algorithm's robustness in attaining a specified level of optimality.

In Theorem 6.2, the regularization coefficient $\lambda$ became bigger as the KL divergence become smaller. This trend aligns intuitively with the principle of relying more heavily on the information provided by the Large Language Model (LLM) when there is evidence that the policy it offers is effective. Essentially, a smaller KL divergence indicates a closer alignment between the LLM's policy and the optimal policy, justifying increased reliance on the LLM's guidance in these scenarios.

**Connection Between the Theoretical Analysis and SLINVIT.** Although SLINVIT differs from the algorithm described in our theoretical analysis in several respects, it is crucial to emphasize that the core objective of our theoretical analysis is to justify the use of log-probability as regularization. Despite the simplifications made for practical implementation, SLINVIT retains log-probability regularization. This adherence ensures that the fundamental element of our analysis is preserved.

## 7 CONCLUSION

Large language models (LLMs) have shown remarkable capabilities in quickly generating viable initial strategies for decision-making tasks, even with minimal or no prior examples. However, these LLM-driven agents struggle to iteratively refine their strategies based on feedback from their environment, mainly because they lack the ability to effectively explore and reason from the feedback. In contrast, reinforcement learning (RL) excels at adapting and improving through feedback. However, it often requires an extensive amount of trial-and-error to gather improvable feedback, hindered by its inability to leverage common sense reasoning. To improve the sample efficiency of RL algorithms, in this work, we propose a novel RL framework with LLM as a policy prior. We prove that the number of samples required by our algorithm is proportional to the KL divergence between the LLM and the optimal policy. This result is further evidenced through experiments in interactive environments such as ALFWorld, InterCode, and BlocksWorld, underscoring our method's improved sample efficiency. For future work, we would like to extend our experiments to more complex and diverse environments would test the scalability and robustness of our framework. Moreover, exploring how various model architectures and pre-training tasks in LLMs influence reinforcement learning performance could provide valuable insights, helping to identify key factors that enhance or hinder learning efficiency and generalization.

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

## A  PROOF OF THEOREM 6.2

*Proof.* To analyze the property of LINVIT, we introduce the definition of the prior-regularized value function. The prior-regularized value function $Q^\pi_{\text{LLM},\lambda,h}$, $V^\pi_{\text{LLM},\lambda,h}(s_h)$ and $V^*_{\text{LLM},\lambda,h}$ are defined as

$$Q^\pi_{\text{LLM},\lambda,h}(s_h,a_h) = r_h(s_h,a_h) + \mathbb{E}_{s_{h+1}\sim P_h}\big[V^\pi_{\text{LLM},\lambda,h+1}(s_{h+1})\big],$$

$$V^\pi_{\text{LLM},\lambda,h}(s_h) = \mathbb{E}_{a_h\sim\pi_h(\cdot|s_h)}Q^\pi_{\text{LLM},\lambda,h}(s_h,a_h) - \lambda\text{KL}\big(\pi_h(\cdot|s_h)\|\pi^{\text{LLM}}_h(\cdot|s_h)\big).$$

$$V^*_{\text{LLM},\lambda,h}(s_h) = \max_\pi V^\pi_{\text{LLM},\lambda,h}(s_h).$$

LINVIT can be viewed as an algorithm that maximizes the prior-regularized value function by interacting with the environment. The prior-regularized value function can be viewed as a regularized version of the original value functions that favors the policies that similar with $\pi^{\text{LLM}}$. The following lemma connects the KL-regularized value with the original value.

**Lemma A.1.** When $\text{KL}(\pi^*\|\pi^{\text{LLM}}) \le \epsilon_{\text{LLM}}$, we have

$$V^*_1(s_1) - V^{\widehat\pi}_1(s_1) \le V^\star_{\text{LLM},\lambda,h}(s_1) - V^{\widehat\pi}_{\text{LLM},\lambda,h}(s_1) + \lambda\epsilon_{\text{LLM}}.$$

Here $\text{KL}(\pi^1\|\pi^2) = \sum_{h=1}^H \mathbb{E}_{\pi^1}[\text{KL}(\pi^1_h(\cdot|s_h)\|\pi^2_h(\cdot|s_h))]$.

*Proof.* See §B.1 for a detailed proof. □

The following lemma show that, our algorithm provably find the optimal policy with respect to the prior-regularized value function with high probability.

**Lemma A.2.** With probability $1-\delta$, we have

$$V^\star_{\text{LLM},\lambda,h}(s_1) - V^{\widehat\pi}_{\text{LLM},\lambda,h}(s_1) \le \epsilon/2.$$

*Proof.* See §B.2 for a detailed proof. □

We denote by $\mathcal{E}_1$ the event in Lemma A.2. We then have $P(\mathcal{E}_1) \ge 1-\delta$. Since we set $\lambda = \epsilon/(2\epsilon_{\text{LLM}})$, we have

$$V^*_1(s_1) - V^{\widehat\pi}_1(s_1) \le V^\star_{\text{LLM},\lambda,h}(s_1) - V^{\widehat\pi}_{\text{LLM},\lambda,h}(s_1) + \epsilon/2 \le \epsilon$$

by Lemma A.1 when we condition on Event $\mathcal{E}_1$. Therefore, we conclude the proof of Theorem 6.2.

□

## B  PROOF OF AUXILIARY LEMMAS

### B.1  PROOF OF LEMMA A.1

*Proof.* By the definitions of $V^{\pi^*}_{\pi^{\text{LLM}},\lambda,h}$ and $V^\star_{\pi^{\text{LLM}},\lambda,h}(s_1)$, we have

$$V^*_1(s_1) = V^{\pi^*}_{\pi^{\text{LLM}},\lambda,h}(s_1) + \lambda\text{KL}(\pi^*,\pi^{\text{LLM}})$$

$$= V^\star_{\pi^{\text{LLM}},\lambda,h}(s_1) + \lambda\text{KL}(\pi^*,\pi^{\text{LLM}}).$$

Therefore, when $\text{KL}(\pi^*,\pi^{\text{LLM}}) \le \epsilon_{\text{LLM}}$, we have

$$V^*_1(s_1) - V^{\widehat\pi}_1(s_1) \le V^\star_{\pi^{\text{LLM}},\lambda,h}(s_1) - V^{\widehat\pi}_{\pi^{\text{LLM}},\lambda,h}(s_1) + \lambda\text{KL}(\pi^*,\pi^{\text{LLM}})$$

$$\le V^\star_{\pi^{\text{LLM}},\lambda,h}(s_1) - V^{\widehat\pi}_{\pi^{\text{LLM}},\lambda,h}(s_1) + \lambda\epsilon_{\text{LLM}},$$

which concludes the proof of Lemma A.1. □

### B.2 PROOF OF LEMMA A.2

*Proof.* The proof of Lemma A.2 consists of two parts. We decompose the regret in the first part, and upper bound the decomposed regret in the second part.

By the definition of $\widehat{\pi}$ in Algorithm LINVIT, we have

$$V^{\star}_{\pi^{\mathrm{LLM}},\lambda,1}(s) - V^{\overline{\pi}^{t+1}}_{\pi^{\mathrm{LLM}},\lambda,1}(s) = \frac{1}{T}\sum_{t=1}^{T}\big[V^{\star}_{\pi^{\mathrm{LLM}},\lambda,1}(s) - V^{\overline{\pi}^{t+1}}_{\pi^{\mathrm{LLM}},\lambda,1}(s)\big]. \tag{B.1}$$

We also have the following lemma.

**Lemma B.1.** We have

$$V^{\star}_{\pi^{\mathrm{LLM}},\lambda,1}(s) - V^{\overline{\pi}^{t+1}}_{\pi^{\mathrm{LLM}},\lambda,1}(s) \le \frac{A}{2\lambda}\sum_{h=1}^{H}\mathbb{E}_{\overline{\pi}^{t+1}}\Big[\max_{a\in\mathcal{A}}\big(\overline{Q}^{t}_{h}(s_h,a) - \underline{Q}^{t}_{h}(s_h,a)\big)^2\Big]$$

holds for all $(t,s) \in [T] \times \mathcal{S}$ with probability $1 - \delta/2$.

*Proof.* See §B.4 for a detailed proof. □

Therefore, we have

$$V^{\star}_{\pi^{\mathrm{LLM}},\lambda,1}(s) - V^{\overline{\pi}^{t+1}}_{\pi^{\mathrm{LLM}},\lambda,1}(s) \le \frac{A}{2T\lambda}\sum_{t=1}^{T}\sum_{h=1}^{H}\mathbb{E}_{\overline{\pi}^{t+1}}\Big[\max_{a\in\mathcal{A}}\big(\overline{Q}^{t}_{h}(s_h,a) - \underline{Q}^{t}_{h}(s_h,a)\big)^2\Big] \tag{B.2}$$

The following lemma upper bounds the decomposed regret.

**Lemma B.2.** We have

$$\sum_{t=1}^{T}\sum_{h=1}^{H}\mathbb{E}_{\overline{\pi}^{t+1}}\Big[\max_{a\in\mathcal{A}}\big(\overline{Q}^{t}_{h}(s_h,a) - \underline{Q}^{t}_{h}(s_h,a)\big)^2\Big] \le 920 SA^3 H^6 \log^2(12HTS^2A/\delta)$$

holds with probability at least $1 - \delta/4$.

*Proof.* See §B.5 for a detailed proof. □

Therefore, we have

$$V^{\star}_{\pi^{\mathrm{LLM}},\lambda,1}(s) - V^{\overline{\pi}^{t+1}}_{\pi^{\mathrm{LLM}},\lambda,1}(s) \le 460 SA^4 H^6 \log^2(12HTS^2A/\delta)/(T\lambda) \tag{B.3}$$

when the event in Lemmas B.1 and B.2 hold. Therefore, when $\lambda = \epsilon/(2\epsilon_{\mathrm{LLM}})$ and $T = CSA^4H^6\log^2(HTSA/\delta)\epsilon_{\mathrm{LLM}}/\epsilon^2$ for some absolute constant $c$, we have $V^{\star}_{\pi^{\mathrm{LLM}},\lambda,1}(s) - V^{\overline{\pi}^{t+1}}_{\pi^{\mathrm{LLM}},\lambda,1}(s) \le \epsilon$, which concludes the proof of Lemma A.1. □

### B.3 PROOF OF LEMMA B.6

*Proof.* First, we have the following lemma, which shows that the bonus we define characterizes the uncertainty of the model estimation with high probability.

**Lemma B.3.** We have

$$\mathbb{1}_{n^t_h(s,a)>0}\,\mathrm{TV}\big(P^*_h(\cdot\mid s,a), P^t_h(\cdot\mid s,a)\big) \le A\sqrt{\frac{\log(4HTS^2A/\delta)}{n^t_h(s,a)}}$$

holds for all $(t,h,s,a) \in [T] \times [H] \times \mathcal{S} \times \mathcal{A}$ with probability at least $1 - \delta/2$.

*Proof.* See §B.6 for a detailed proof. □

We denote by $\mathcal{E}_3$ the event in Lemma B.3. In the following part of the proof, we condition on Event $\mathcal{E}_3$. We prove Lemma B.6 using induction on $h$. By the definition of $\overline{Q}_{H+1}^t$ and $Q_{\mathrm{LLM},\lambda,H+1}^\star(s,a)$, Lemma B.6 holds when $h = H+1$. We also have the following lemma, which shows that the prior-regularized value function is bounded.

**Lemma B.4** (Boundedness of $Q_{\mathrm{LLM},\lambda,h}^\star$). We have

$$0 \le Q_{\mathrm{LLM},\lambda,h}^\star(s,a) \le H+1-h; \qquad 0 \le V_{\mathrm{LLM},\lambda,h}^\star(s) \le H+1-h$$

holds for all $(s,a,h) \in \mathcal{S} \times \mathcal{A} \times [H+1]$.

*Proof.* See §B.7 for a detailed proof. $\qquad\square$

By Lemma B.4, Lemma B.6 obviously holds when $\overline{Q}_h^t \ge H$. Otherwise, we have

$$\overline{Q}_h^t(s,a) - Q_{\mathrm{LLM},\lambda,h}^\star(s,a) = \sum_{s'\in\mathcal{S}} P_h^t(s'\mid s,a)\overline{V}_{h+1}^t(s') - \sum_{s'\in\mathcal{S}} P_h^*(s'\mid s,a)V_{\mathrm{LLM},\lambda,h+1}^\star(s') + u_h^t(s,a)$$

$$\ge \sum_{s'\in\mathcal{S}} \left[P_h^t(s'\mid s,a) - P_h^*(s'\mid s,a)\right]V_{\mathrm{LLM},\lambda,h+1}^\star(s') + u_h^t(s,a)$$

when the induction hypothesis holds. Since $|V_{\mathrm{LLM},\lambda,h+1}^\star(s')| \le H$, we have

$$\overline{Q}_h^t(s,a) - Q_{\mathrm{LLM},\lambda,h}^\star(s,a) \ge u_h^t(s,a) - H \sum_{s'\in\mathcal{S}} \left|P_h^t(s'\mid s,a) - P_h^*(s'\mid s,a)\right|$$

$$= u_h^t(s,a) - H\,\mathrm{TV}(P_h^t(\cdot\mid s,a), P_h^*(\cdot\mid s,a)).$$

When we condition on $\mathcal{E}_3$, we have

$$H\,\mathrm{TV}(P_h^t(\cdot\mid s,a), P_h^*(\cdot\mid s,a)) \le \max\left\{2H, \sqrt{\frac{\log(4HTS^2A/\delta)}{n_h^t(s,a)}}\right\} = u_h^t(s,a), \qquad (\mathrm{B.4})$$

which implies $\overline{Q}_h^t(s,a) \ge Q_{\mathrm{LLM},\lambda,h}^\star(s,a)$. Therefore, we have

$$\overline{V}_h^t(s) = \max_{\pi(\cdot\mid s)} \sum_{a\in\mathcal{A}} \pi(a\mid s)\overline{Q}_h^t(s,a) - \lambda\mathrm{KL}\big(\pi(\cdot\mid s), \pi^{\mathrm{LLM}}(\cdot\mid s)\big)$$

$$\ge \max_{\pi(\cdot\mid s)} \sum_{a\in\mathcal{A}} \pi(a\mid s)Q_{\mathrm{LLM},\lambda,h}^\star(s,a) - \lambda\mathrm{KL}\big(\pi(\cdot\mid s), \pi^{\mathrm{LLM}}(\cdot\mid s)\big) = V_{\mathrm{LLM},\lambda,h}^\star(s),$$

which concludes the first part of the proof.

By the definition of $\underline{Q}_{H+1}^t$ and $Q_{\mathrm{LLM},\lambda,H+1}^\star(s,a)$, Lemma B.6 holds when $h = H+1$. By the boundedness of $Q_{\mathrm{LLM},\lambda,h}^\star$, Lemma B.6 obviously holds when $\underline{Q}_h^t \le 0$. Otherwise, we have

$$\underline{Q}_h^t(s,a) - Q_{\mathrm{LLM},\lambda,h}^\star(s,a) \le \sum_{s'\in\mathcal{S}} P_h^t(s'\mid s,a)\underline{V}_{h+1}^t(s') - \sum_{s'\in\mathcal{S}} P_h^*(s'\mid s,a)V_{\mathrm{LLM},\lambda,h+1}^\star(s') - u_h^t(s,a)$$

$$\le \sum_{s'\in\mathcal{S}} \left[P_h^t(s'\mid s,a) - P_h^*(s'\mid s,a)\right]V_{\mathrm{LLM},\lambda,h+1}^\star(s') - u_h^t(s,a)$$

when the induction hypothesis holds. Since $|V_{\mathrm{LLM},\lambda,h+1}^\star(s')| \le H$, we have

$$\underline{Q}_h^t(s,a) - Q_{\mathrm{LLM},\lambda,h}^\star(s,a) \ge H \sum_{s'\in\mathcal{S}} \left|P_h^t(s'\mid s,a) - P_h^*(s'\mid s,a)\right| - u_h^t(s,a)$$

$$= H\,\mathrm{TV}(P_h^t(\cdot\mid s,a), P_h^*(\cdot\mid s,a)) - u_h^t(s,a).$$

Therefore, we have $\underline{Q}_h^t(s,a) \ge Q_{\mathrm{LLM},\lambda,h}^\star(s,a)$ when we condition on $\mathcal{E}_3$ by (B.4). We have

$$\underline{V}_h^t(s) = \max_{\pi(\cdot\mid s)} \sum_{a\in\mathcal{A}} \pi(a\mid s)\underline{Q}_h^t(s,a) - \lambda\mathrm{KL}\big(\pi(\cdot\mid s), \pi^{\mathrm{LLM}}(\cdot\mid s)\big)$$

$$\ge \max_{\pi(\cdot\mid s)} \sum_{a\in\mathcal{A}} \pi(a\mid s)Q_{\mathrm{LLM},\lambda,h}^\star(s,a) - \lambda\mathrm{KL}\big(\pi(\cdot\mid s), \pi^{\mathrm{LLM}}(\cdot\mid s)\big) = V_{\mathrm{LLM},\lambda,h}^\star(s),$$

which conclude the proof of Lemma B.6. $\qquad\square$

### B.4 PROOF OF LEMMA B.1

*Proof.* We have the following lemma.

**Lemma B.5.** For two vectors $x = (x_1, \ldots, x_A) \in \mathbb{R}^A$ and $\bar{x} = (\bar{x}_1, \ldots, \bar{x}_A) \in \mathbb{R}^A$, we have

$$\max_{\sum_{i=1}^A \beta_i = 1, \beta_i > 0} \left[ \sum_{i=1}^A x_i \beta_i - \lambda \sum_{i=1}^A \beta_i \log \frac{\beta_i}{\widetilde{\beta}_i} \right] - \max_{\sum_{i=1}^A \beta_i = 1, \beta_i > 0} \left[ \sum_{i=1}^A \bar{x}_i \beta_i - \lambda \sum_{i=1}^A \beta_i \log \frac{\beta_i}{\widetilde{\beta}_i} \right]$$

$$\leq \sum_{i=1}^A \bar{\beta}_i (x_i - \bar{x}_i) + \frac{A}{2\lambda} \max_i |x_i - \bar{x}_i|^2,$$

where $\{\bar{\beta}_i\} = \mathrm{argmax}_{\sum_{i=1}^A \beta_i = 1, \beta_i > 0} \left[ \sum_{i=1}^A \bar{x}_i \beta_i - \sum_{i=1}^A \beta_i \log \frac{\beta_i}{\widetilde{\beta}_i} \right].$

*Proof.* See §B.8 for a detailed proof. □

We prove Lemma B.1 using induction on $h$. The statement obviously holds when $h = H + 1$.

$$V^\star_{\pi^{\mathrm{LLM}}, \lambda, h}(s) - V^{\bar{\pi}^{t+1}}_{\pi^{\mathrm{LLM}}, \lambda, h}(s) = V^\star_{\pi^{\mathrm{LLM}}, \lambda, h}(s) - \overline{V}^t_h(s) + \overline{V}^t_h(s) - V^{\bar{\pi}^{t+1}}_{\pi^{\mathrm{LLM}}, \lambda, h}(s). \tag{B.5}$$

By the definition of $V^\star_{\pi^{\mathrm{LLM}}, \lambda, h}$, $V^{\bar{\pi}^{t+1}}_{\pi^{\mathrm{LLM}}, \lambda, h}$ and Lemma B.5, we have

$$V^\star_{\pi^{\mathrm{LLM}}, \lambda, h}(s) - \overline{V}^t_h(s) \tag{B.6}$$

$$\leq \sum_{a \in \mathcal{A}} \bar{\pi}^{t+1}_h(a \mid s) \left[ Q^\star_{\pi^{\mathrm{LLM}}, \lambda, h}(s, a) - \overline{Q}^t_h(s, a) \right] + \frac{A}{2\lambda} \max_a |Q^\star_{\pi^{\mathrm{LLM}}, \lambda, h}(s, a) - \overline{Q}^t_h(s, a)|^2.$$

We also have

$$\overline{V}^t_h(s) - V^{\bar{\pi}^{t+1}}_{\pi^{\mathrm{LLM}}, \lambda, h}(s) = \sum_{a \in \mathcal{A}} \bar{\pi}^{t+1}_h(a \mid s) \left[ \overline{Q}^t_h(s, a) - Q^{\bar{\pi}^{t+1}}_{\pi^{\mathrm{LLM}}, \lambda, h}(s, a) \right]. \tag{B.7}$$

Combining (B.5), (B.6) and (B.7), we have

$$V^\star_{\pi^{\mathrm{LLM}}, \lambda, h}(s) - V^{\bar{\pi}^{t+1}}_{\pi^{\mathrm{LLM}}, \lambda, h}(s)$$

$$\leq \sum_{a \in \mathcal{A}} \bar{\pi}^{t+1}_h(a \mid s) \left[ Q^\star_{\pi^{\mathrm{LLM}}, \lambda, h}(s, a) - Q^{\bar{\pi}^{t+1}}_{\pi^{\mathrm{LLM}}, \lambda, h}(s, a) \right] + \frac{A}{2\lambda} \max_a |Q^\star_{\pi^{\mathrm{LLM}}, \lambda, h}(s, a) - \overline{Q}^t_h(s, a)|^2$$

$$= \mathbb{E}_{\bar{\pi}^{t+1}} \left[ V^\star_{\pi^{\mathrm{LLM}}, \lambda, h+1}(s_{h+1}, a_{h+1}) - V^{\bar{\pi}^{t+1}}_{\pi^{\mathrm{LLM}}, \lambda, h+1}(s_{h+1}, a_{h+1}) \mid s_h = s \right] + \frac{A}{2\lambda} \max_a |Q^\star_{\pi^{\mathrm{LLM}}, \lambda, h}(s, a) - \overline{Q}^t_h(s, a)|^2$$

By induction, we easily have

$$V^\star_{\pi^{\mathrm{LLM}}, \lambda, h}(s) - V^{\bar{\pi}^{t+1}}_{\pi^{\mathrm{LLM}}, \lambda, h}(s) \leq \frac{A}{2\lambda} \sum_{h'=h}^H \mathbb{E}_{\bar{\pi}^{t+1}} \left[ \max_a |Q^\star_{\pi^{\mathrm{LLM}}, \lambda, h'}(s_{h'}, a_{h'}) - \overline{Q}^t_{h'}(s_{h'}, a_{h'})|^2 \mid s_h = s \right].$$

$$\tag{B.8}$$

We also have the following lemma, which shows that $Q^\star_{\pi^{\mathrm{LLM}}, \lambda, h'}$ lies between $\overline{Q}^t_{h'}$ and $\underline{Q}^t_{h'}$.

**Lemma B.6.** We have

$$\underline{Q}^t_h(s, a) \leq Q^\star_{\pi^{\mathrm{LLM}}, \lambda, h}(s, a) \leq \overline{Q}^t_h(s, a), \qquad \underline{V}^t_{\lambda, h}(s) \leq V^\star_{\pi^{\mathrm{LLM}}, \lambda, h}(s) \leq \overline{V}^t_h(s)$$

holds for all $t \in \mathbb{N}$, $(h, s, a) \in [H] \times \mathcal{S} \times \mathcal{A}$ with probability $1 - \delta/2$.

*Proof.* See §B.3 for a detailed proof. □

Therefore, by (B.8), we have

$$V^\star_{\pi^{\mathrm{LLM}}, \lambda, h}(s) - V^{\bar{\pi}^{t+1}}_{\pi^{\mathrm{LLM}}, \lambda, h}(s) \leq \frac{A}{2\lambda} \sum_{h'=h}^H \mathbb{E}_{\bar{\pi}^{t+1}} \left[ \max_a |\overline{Q}^t_{h'}(s_{h'}, a_{h'}) - \underline{Q}^t_{h'}(s_{h'}, a_{h'})|^2 \mid s_h = s \right]$$

$$\tag{B.9}$$

when the event in Lemma B.6 holds, which conclude the proof of Lemma B.1.

□

### B.5 Proof of Lemma B.2

*Proof.* By the definition of $\pi^t$ in (3.4), we have

$$
\mathbb{E}_{\overline{\pi}^{t+1}}\left[\max_{a\in\mathcal{A}}\left(\overline{Q}_h^t(s_h,a) - \underline{Q}_h^t(s_h,a)\right)^2\right] \le H\left(\frac{H}{H-1}\right)^{h-1}\mathbb{E}_{\pi^{t+1}}\left[\left(\overline{Q}_h^t(s_h,a_h) - \underline{Q}_h^t(s_h,a_h)\right)^2\right]
$$

(B.10)

$$
\le eH\mathbb{E}_{\pi^{t+1}}\left[\left(\overline{Q}_h^t(s_h,a_h) - \underline{Q}_h^t(s_h,a_h)\right)^2\right].
$$

Next we analyze the expression under the square. First, we have

$$
\overline{Q}_h^t(s_h,a_h) - \underline{Q}_h^t(s_h,a_h) \le 2u_h^t(s_h,a_h) + \sum_{a\in\mathcal{A}} P_h^t(s'|s_h,a_h)[\overline{V}_{h+1}^t(s') - \underline{V}_{h+1}^t(s')]
$$

$$
\le 2u_h^t(s_h,a_h) + \sum_{a\in\mathcal{A}} P_h^*(s'|s_h,a_h)[\overline{V}_{h+1}^t(s') - \underline{V}_{h+1}^t(s')] + H\,\mathrm{TV}(P_h^t(\cdot\mid s,a), P_h^*(\cdot\mid s,a)).
$$

Therefore, by (B.4), we have

$$
\overline{Q}_h^t(s_h,a_h) - \underline{Q}_h^t(s_h,a_h) \le 3u_h^t(s_h,a_h) + \sum_{a\in\mathcal{A}} P_h^*(s'|s_h,a_h)[\overline{V}_{h+1}^t(s') - \underline{V}_{h+1}^t(s')].
$$

Therefore, we have

$$
\overline{Q}_h^t(s_h,a_h) - \underline{Q}_h^t(s_h,a_h) \le 3AH\cdot\mathbb{E}_{\overline{\pi}^{t+1}}\left[\sum_{h'=h}^{H}\sqrt{\frac{\log(4HTS^2A/\delta)}{n_h^t(s,a)}}\bigg|s_h\right]
$$

$$
\le 9AH\cdot\mathbb{E}_{\pi^{t+1}}\left[\sum_{h'=h}^{H}\sqrt{\frac{\log(4HTS^2A/\delta)}{n_h^t(s,a)}}\bigg|s_h\right]
$$

by induction and the definition of $u_h^t$ in (3.1). By Cauchy inequality, we have

$$
\overline{Q}_h^t(s_h,a_h) - \underline{Q}_h^t(s_h,a_h) \le 9AH^{3/2}\sqrt{\mathbb{E}_{\pi^{t+1}}\left[\sum_{h'=h}^{H}\frac{\log(4HTS^2A/\delta)}{n_h^t(s,a)}\bigg|s_h\right]}.
$$

(B.11)

Combining (B.10) and (B.11), we have

$$
\mathbb{E}_{\overline{\pi}^{t+1}}\left[\max_{a\in\mathcal{A}}\left(\overline{Q}_h^t(s_h,a) - \underline{Q}_h^t(s_h,a)\right)^2\right] \le 230A^2H^4\mathbb{E}_{\pi^{t+1}}\left[\mathbb{E}_{\pi^{t+1}}\left[\sum_{h'=h}^{H}\frac{\log(4HTS^2A/\delta)}{n_h^t(s,a)}\bigg|s_h\right]\right]
$$

(B.12)

$$
= 230A^2H^4\mathbb{E}_{\pi^{t+1}}\left[\sum_{h'=h}^{H}\frac{\log(4HTS^2A/\delta)}{n_h^t(s,a)}\bigg|s_h\right].
$$

We also have the following lemma.

**Lemma B.7.** We define

$$
\mathcal{E}^{\mathrm{cnt}} \triangleq \{n_h^t(s,a) \ge \frac{1}{2}\bar{n}_h^t(s,a) - \log(12SAH/\delta) \text{ for all } (s,a,h,t)\in\mathcal{S}\times\mathcal{A}\times[H]\times[T]\},
$$

Then we have $P(\mathcal{E}^{\mathrm{cnt}}) \ge 1 - \delta/4$.

*Proof.* This is Lemma 3 of Ménard et al. (2021). See Ménard et al. (2021) for a detailed proof. □

When we condition on $\mathcal{E}^{\mathrm{cnt}}$, we have

$$
\frac{\log(4HTS^2A/\delta)}{n_h^t(s,a)} \le \frac{\log(4HTS^2A/\delta) + \log(12SAH/\delta)}{n_h^t(s,a) + \log(12SAH/\delta)} \le \frac{2\log(12HTS^2A/\delta)}{n_h^t(s,a) + \log(12SAH/\delta)}
$$

$$
\le \frac{4\log(12HTS^2A/\delta)}{\bar{n}_h^t(s,a)}.
$$

Therefore, by (B.12), we have

$$
\mathbb{E}_{\overline{\pi}^{t+1}} \left[ \max_{a \in \mathcal{A}} \left( \overline{Q}_h^t(s_h, a) - \underline{Q}_h^t(s_h, a) \right)^2 \right] \le 920 A^2 H^4 \mathbb{E}_{\pi^{t+1}} \left[ \sum_{h'=h}^{H} \frac{\log(12HTS^2A/\delta)}{\bar{n}_h^t(s, a)} \middle| s_h \right].
$$

Therefore, we have

$$
\begin{aligned}
\sum_{t=1}^{T} \sum_{h=1}^{H} \mathbb{E}_{\overline{\pi}^{t+1}} \left[ \max_{a \in \mathcal{A}} \left( \overline{Q}_h^t(s_h, a) - \underline{Q}_h^t(s_h, a) \right)^2 \right] &\le \sum_{t=1}^{T} \sum_{h=1}^{H} 920 A^2 H^4 \mathbb{E}_{\pi^{t+1}} \left[ \sum_{h'=h}^{H} \frac{\log(12HTS^2A/\delta)}{\bar{n}_{h'}^t(s, a)} \middle| s_h \right] \\
&\le \sum_{t=1}^{T} \sum_{h=1}^{H} 920 A^2 H^5 \mathbb{E}_{\pi^{t+1}} \left[ \frac{\log(12HTS^2A/\delta)}{\bar{n}_h^t(s, a)} \middle| s_h \right] \\
&\le 920 S A^3 H^6 \log(12HTS^2A/\delta) \log T,
\end{aligned}
$$

which concludes the proof of Lemma B.2. $\qquad\square$

## B.6  PROOF OF LEMMA B.3

*Proof.* We denote by $n_h^t(s, a)$ the number of times the state action-pair $(s, a)$ was visited in step $h$ in the first $t$ episodes, $n_h^t(s, a, s')$ the number of times the state action next state -pair $(s, a, s')$ was visited in step $h$ in the first $t$ episodes, and define $N_h^t(s, a, s') = n_h^t(s, a) P_h(s' \mid s, a)$. By Hoeffding's inequality, we have

$$
\mathbb{1}_{n_h^t(s,a)>0} \left| \frac{n_h^t(s, a, s') - N_h^t(s, a, s')}{\sqrt{n_h^t(s, a)}} \right| \le \sqrt{\log(4HTS^2A/\delta)} \tag{B.13}
$$

holds for a fix $(t, h, s, a, s') \in [T] \times [H] \times \mathcal{S} \times \mathcal{A} \times \mathcal{S}$ with probability at least $1 - \delta/(2HTS^2A)$. By taking a union bound over all $(t, h, s, a, s') \in [T] \times [H] \times \mathcal{S} \times \mathcal{A} \times \mathcal{S}$, we have (B.13) holds for all $(t, h, s, a, s') \in [T] \times [H] \times \mathcal{S} \times \mathcal{A} \times \mathcal{S}$ with probability at least $1 - \delta/2$. We denote by $\mathcal{E}_2$ that (B.13) holds for all $(t, h, s, a, s') \in [T] \times [H] \times \mathcal{S} \times \mathcal{A} \times \mathcal{S}$. When condition on $\mathcal{E}_2$, we have

$$
\begin{aligned}
\mathbb{1}_{n_h^t(s,a)>0} \operatorname{TV}\left( P_h^*(\cdot \mid s, a), P_h^t(\cdot \mid s, a) \right) &= \sum_{s' \in \mathcal{S}} \mathbb{1}_{n_h^t(s,a)>0} \left| \frac{n_h^t(s, a, s') - N_h^t(s, a, s')}{n_h^t(s, a)} \right| \\
&\le A \sqrt{\frac{\log(4HTS^2A/\delta)}{n_h^t(s, a)}}.
\end{aligned}
$$

We conclude the proof by noticing that $P(\mathcal{E}_2) \ge 1 - \delta/2$. $\qquad\square$

## B.7  PROOF OF LEMMA B.4

*Proof.* We prove this using induction. Lemma B.4 obviously holds when $h = H + 1$. If Lemma B.4 hold for $h + 1$, we have

$$
Q_{\mathrm{LLM},\lambda,h}^\star(s, a) = r_h(s, a) + \sum_{s' \in \mathcal{S}} P_h^*(s'|s, a) V_{\mathrm{LLM},\lambda,h+1}^\star(s, a) \ge 0.
$$

We also have

$$
Q_{\mathrm{LLM},\lambda,h}^\star(s, a) = r_h(s, a) + \sum_{s' \in \mathcal{S}} P_h^*(s'|s, a) V_{\mathrm{LLM},\lambda,h+1}^\star(s, a) \le 1 + \sum_{s' \in \mathcal{S}} P_h^*(s'|s, a)(H - h) = H - h + 1.
$$

Using the property of constraint optimization, we have

$$
V_{\mathrm{LLM},\lambda,h}^\star(s) = \lambda \log \Big( \sum_{a \in \mathcal{A}} \pi_h^{\mathrm{LLM}}(a|s) \exp(Q_{\mathrm{LLM},\lambda,h}^\star(s, a)/\lambda) \Big).
$$

since $0 \le Q_{\mathrm{LLM},\lambda,h}^\star(s, a) \le H + 1 - h$, we have $0 \le V_{\mathrm{LLM},\lambda,h}^\star(s) \le H + 1 - h$. Therefore, we conclude the proof of Lemma B.4 using induction. $\qquad\square$

## B.8 PROOF OF LEMMA B.5

*Proof.* For $\boldsymbol{x} = (x_1, \ldots, x_A)^\top$ and $\boldsymbol{\beta} = (\beta_1, \ldots, \beta_A)^\top$, we define

$$f(\boldsymbol{x}, \boldsymbol{\beta}) = \boldsymbol{x}^\top \boldsymbol{\beta} - \lambda \sum_{i=1}^A \beta_i \log \frac{\beta_i}{\widetilde{\beta}_i}, \quad f^*(\boldsymbol{x}) = \max_{\sum_{i=1}^A \beta_i = 1, \beta_i > 0} f(\boldsymbol{x}, \boldsymbol{\beta}), \quad \boldsymbol{\beta}^*(x) = \operatorname*{argmax}_{\sum_{i=1}^A \beta_i = 1, \beta_i > 0} f(\boldsymbol{x}, \boldsymbol{\beta}). \tag{B.14}$$

We have the following lemma.

**Lemma B.8.** We have

$$f^*(\boldsymbol{x}) = \lambda \log \Big[ \sum_{i=1}^A \widetilde{\beta}_i \exp(x_i/\lambda) \Big], \nabla f^*(\boldsymbol{x}) = \boldsymbol{\beta}^*(\boldsymbol{x}), \beta_i(\boldsymbol{x}) = \frac{\partial}{\partial x_i} f^*(\boldsymbol{x}) = \frac{\widetilde{\beta}_i \exp(x_i/\lambda)}{\sum_{j=1}^A \widetilde{\beta}_j \exp(x_j/\lambda)}.$$

Moreover, we have $\|\nabla f^*(\boldsymbol{x}_1) - \nabla f^*(\boldsymbol{x}_2)\|_1 \le \frac{A}{\lambda} \|\boldsymbol{x}_1 - \boldsymbol{x}_2\|_\infty$.

*Proof.* See §B.9 for a detailed proof. □

First, we have

$$f^*(\boldsymbol{x}) - f^*(\bar{\boldsymbol{x}}) = \nabla f^*(\bar{\boldsymbol{x}})(\boldsymbol{x} - \bar{\boldsymbol{x}}) + + \int_0^1 \Big( \nabla f^*\big(t\boldsymbol{x} + (1-t)\bar{\boldsymbol{x}}\big) - \nabla f^*(\bar{\boldsymbol{x}}) \Big)^\top (\boldsymbol{x} - \bar{\boldsymbol{x}}) \mathrm{d}t$$

$$\le \nabla f^*(\bar{\boldsymbol{x}})(\boldsymbol{x} - \bar{\boldsymbol{x}}) + + \int_0^1 \big\| \nabla f^*\big(t\boldsymbol{x} + (1-t)\bar{\boldsymbol{x}}\big) - \nabla f^*(\bar{\boldsymbol{x}}) \big\|_1 \|\boldsymbol{x} - \bar{\boldsymbol{x}}\|_\infty \mathrm{d}t.$$

By Lemma B.8, we have

$$f^*(\boldsymbol{x}) - f^*(\bar{\boldsymbol{x}}) \le \nabla f^*(\bar{\boldsymbol{x}})(\boldsymbol{x} - \bar{\boldsymbol{x}}) + \frac{A}{\lambda} \int_0^1 \big\| t\boldsymbol{x} + (1-t)\bar{\boldsymbol{x}} - \bar{\boldsymbol{x}} \big\|_\infty \|\boldsymbol{x} - \bar{\boldsymbol{x}}\|_\infty \mathrm{d}t$$

$$= \nabla f^*(\bar{\boldsymbol{x}})(\boldsymbol{x} - \bar{\boldsymbol{x}}) + \frac{A}{2\lambda} \|\boldsymbol{x} - \bar{\boldsymbol{x}}\|_\infty^2 \tag{B.15}$$

We conclude the proof of Lemma B.5 by combining (B.15) with Lemma B.8. □

## B.9 PROOF OF LEMMA B.8

*Proof.* By the property of constraint optimization, we have $\frac{\partial}{\partial \beta} f\big(\boldsymbol{x}, \boldsymbol{\beta}(\boldsymbol{x})\big) = c(1, \ldots, 1)^\top$ for some constant $c$. By direct computation, we have

$$\frac{\partial}{\partial \beta} f\big(\boldsymbol{x}, \boldsymbol{\beta}(\boldsymbol{x})\big) = x - \lambda \log \boldsymbol{\beta} + \lambda \log \widetilde{\boldsymbol{\beta}} - \lambda(1, \ldots, 1)^\top,$$

which implies $\beta_i(\boldsymbol{x}) \propto \widetilde{\beta}_i \exp(x_i/\lambda)$. Since $\sum_{i=1}^A \beta_i(\boldsymbol{x} \propto) = 1$, we have

$$\beta_i(\boldsymbol{x}) = \frac{\widetilde{\beta}_i \exp(x_i/\lambda)}{\sum_{j=1}^A \widetilde{\beta}_j \exp(x_j/\lambda)}. \tag{B.16}$$

By the definition of $f^*$, we have

$$f^*(\boldsymbol{x}) = f(\boldsymbol{x}, \boldsymbol{\beta}_i(\boldsymbol{x})) = \frac{\sum_{i=1}^A x_i \widetilde{\beta}_i \exp(x_i/\lambda)}{\sum_{i=1}^A \widetilde{\beta}_i \exp(x_i/\lambda)} - \frac{\sum_{i=1}^A x_i \widetilde{\beta}_i \exp(x_i/\lambda)}{\sum_{i=1}^A \widetilde{\beta}_i \exp(x_i/\lambda)} + \lambda \log \big( \sum_{i=1}^A \widetilde{\beta}_i \exp(x_i/\lambda) \big)$$

$$= \lambda \log \big( \sum_{i=1}^A \widetilde{\beta}_i \exp(x_i/\lambda) \big). \tag{B.17}$$

Combining (B.16) and (B.17), we have $\nabla f^*(\boldsymbol{x}) = \boldsymbol{\beta}(\boldsymbol{x})$. We define $\Delta = \{\boldsymbol{\beta} \mid \sum_{i=1}^A \beta_i = 1, \beta_i > 0\}$, we have the following lemma.

**Lemma B.9.** For any $\boldsymbol{x}, \boldsymbol{\beta} \in \Delta$, we have

$$\lambda \sum_{i=1}^{A} \beta_i \log \frac{\beta_i}{\widetilde{\beta}_i} \geq \lambda \sum_{i=1}^{A} \beta_i(\boldsymbol{x}) \log \frac{\beta_i(\boldsymbol{x})}{\widetilde{\beta}_i} + \boldsymbol{x}^{\top} \big(\boldsymbol{\beta} - \boldsymbol{\beta}(\boldsymbol{x})\big) + \frac{\lambda}{2A} \|\boldsymbol{\beta} - \boldsymbol{\beta}(\boldsymbol{x})\|_1^2,$$

where $\boldsymbol{\beta}(\boldsymbol{x})$ is defined in (B.14).

*Proof.* See §B.10 for a detailed proof. □

By Lemma B.9, we have

$$\lambda \sum_{i=1}^{A} \beta_i(\boldsymbol{x}_2) \log \frac{\beta_i(\boldsymbol{x}_2)}{\widetilde{\beta}_i} \geq \lambda \sum_{i=1}^{A} \beta_i(\boldsymbol{x}_1) \log \frac{\beta_i(\boldsymbol{x}_1)}{\widetilde{\beta}_i} + \boldsymbol{x}_1^{\top} \big(\boldsymbol{\beta}(\boldsymbol{x}_2) - \boldsymbol{\beta}(\boldsymbol{x}_1)\big) + \frac{\lambda}{2A} \|\boldsymbol{\beta}(\boldsymbol{x}_2) - \boldsymbol{\beta}(\boldsymbol{x}_1)\|_1^2,$$

$$\lambda \sum_{i=1}^{A} \beta_i(\boldsymbol{x}_1) \log \frac{\beta_i(\boldsymbol{x}_1)}{\widetilde{\beta}_i} \geq \lambda \sum_{i=1}^{A} \beta_i(\boldsymbol{x}_2) \log \frac{\beta_i(\boldsymbol{x}_2)}{\widetilde{\beta}_i} + \boldsymbol{x}_2^{\top} \big(\boldsymbol{\beta}(\boldsymbol{x}_1) - \boldsymbol{\beta}(\boldsymbol{x}_2)\big) + \frac{\lambda}{2A} \|\boldsymbol{\beta}(\boldsymbol{x}_2) - \boldsymbol{\beta}(\boldsymbol{x}_1)\|_1^2,$$

which implies

$$\frac{\lambda}{A} \|\boldsymbol{\beta}(\boldsymbol{x}_2) - \boldsymbol{\beta}(\boldsymbol{x}_1)\|_1^2 \leq (\boldsymbol{x}_1 - \boldsymbol{x}_2)^{\top} \big(\boldsymbol{\beta}(\boldsymbol{x}_1) - \boldsymbol{\beta}(\boldsymbol{x}_2)\big) \leq \|\boldsymbol{\beta}(\boldsymbol{x}_2) - \boldsymbol{\beta}(\boldsymbol{x}_1)\|_1 \|\boldsymbol{x}_1 - \boldsymbol{x}_2\|_\infty.$$

Therefore, we have

$$\frac{\lambda}{A} \|\boldsymbol{\beta}(\boldsymbol{x}_2) - \boldsymbol{\beta}(\boldsymbol{x}_1)\|_1 \leq \|\boldsymbol{x}_1 - \boldsymbol{x}_2\|_\infty,$$

which concludes the proof of Lemma B.8 □

### B.10 PROOF OF LEMMA B.9

*Proof.* Let $\beta_i$ be the $i$-th element of $\boldsymbol{\beta}$, and $\beta_i(\boldsymbol{x})$ be the $i$-th element of $\boldsymbol{\beta}$. We define $g(\boldsymbol{\beta}) = \lambda \sum_{i=1}^{A} \beta_i \log \frac{\beta_i}{\widetilde{\beta}_i}$. By the property of the function $g$, we have

$$g(\boldsymbol{\beta}) - g\big(\boldsymbol{\beta}(\boldsymbol{x})\big) = \nabla g\big(\boldsymbol{\beta}(\boldsymbol{x})\big)^{\top} \big(\boldsymbol{\beta} - \boldsymbol{\beta}(\boldsymbol{x})\big) + \sum_{i=1}^{A} \int_{\beta_i(\boldsymbol{x})}^{\beta_i} \int_{\beta_i(\boldsymbol{x})}^{u} \frac{\lambda}{v} \mathrm{d}v \mathrm{d}u.$$

Since $\beta_i < 1$, we have

$$g(\boldsymbol{\beta}) - g\big(\boldsymbol{\beta}(\boldsymbol{x})\big) \geq \nabla g\big(\boldsymbol{\beta}(\boldsymbol{x})\big)^{\top} \big(\boldsymbol{\beta} - \boldsymbol{\beta}(\boldsymbol{x})\big) + \lambda \sum_{i=1}^{A} \int_{\beta_i(\boldsymbol{x})}^{\beta_i} \int_{\beta_i(\boldsymbol{x})}^{u} 1 \mathrm{d}v \mathrm{d}u \tag{B.18}$$

$$= \nabla g\big(\boldsymbol{\beta}(\boldsymbol{x})\big)^{\top} \big(\boldsymbol{\beta} - \boldsymbol{\beta}(\boldsymbol{x})\big) + \frac{\lambda}{2} \sum_{i=1}^{A} |\beta_i(\boldsymbol{x}) - \beta_i|^2$$

$$\geq \nabla g\big(\boldsymbol{\beta}(\boldsymbol{x})\big)^{\top} \big(\boldsymbol{\beta} - \boldsymbol{\beta}(\boldsymbol{x})\big) + \frac{\lambda}{2A} \|\boldsymbol{\beta}(\boldsymbol{x}) - \boldsymbol{\beta}\|_1^2.$$

By the definition of the function $g$, we have

$$\nabla g\big(\boldsymbol{\beta}(\boldsymbol{x})\big) = (\lambda \log \beta_1 + \lambda - \lambda \log \widetilde{\beta}_1, \ldots, \lambda \log \beta_A + \lambda - \lambda \log \widetilde{\beta}_A)^{\top}. \tag{B.19}$$

By the definition of $\boldsymbol{\beta}(\boldsymbol{x})$ in (B.14), we can easily obtain

$$\beta_i(\boldsymbol{x}) = \frac{\widetilde{\beta}_i \exp(x_i/\lambda)}{\sum_{i=1}^{A} \widetilde{\beta}_i \exp(x_i/\lambda)}. \tag{B.20}$$

Combining (B.19) with (B.20), we have

$$\nabla g\big(\boldsymbol{\beta}(\boldsymbol{x})\big) = \boldsymbol{x} + \lambda \Big(1 - \log\big(\sum_{i=1}^{A} \widetilde{\beta}_i \exp(x_i/\lambda)\big)\Big)(1, 1, \ldots, 1)^{\top}.$$

Therefore, we have

$$\big(\nabla g\big(\boldsymbol{\beta}(\boldsymbol{x})\big) - \boldsymbol{x}\big)^\top \big(\boldsymbol{\beta} - \boldsymbol{\beta}(\boldsymbol{x})\big)$$

$$= \lambda\Big(1 - \log\big(\sum_{i=1}^{A} \widetilde{\beta}_i \exp(x_i/\lambda)\big)\Big)(1, 1, \ldots, 1)^\top \boldsymbol{\beta} - \lambda\Big(1 - \log\big(\sum_{i=1}^{A} \widetilde{\beta}_i \exp(x_i/\lambda)\big)\Big)(1, 1, \ldots, 1)^\top \boldsymbol{\beta}(\boldsymbol{x}).$$

By the definition of the region $\Delta$ in Lemma B.8, we have $(1, 1, \ldots, 1)^\top \boldsymbol{\beta} = (1, 1, \ldots, 1)^\top \boldsymbol{\beta}(\boldsymbol{x}) = 1$. Therefore, we have

$$\big(\nabla g\big(\boldsymbol{\beta}(\boldsymbol{x})\big) - \boldsymbol{x}\big)^\top \big(\boldsymbol{\beta} - \boldsymbol{\beta}(\boldsymbol{x})\big) = 0. \tag{B.21}$$

We conclude the proof of Lemma B.9 by combining (B.18) and (B.21). □

## C EXPERIMENT DETAILS

We first describe how the rule-based value estimator is implemented. For the "put" task, we prompt the LLM with the following prompt:

```
                              ─── Input ───

 Given the following successful trajectory, output sub-goals that are
 needed to complete the task ``put A on B" and write code to determine
 how many sub-goals are satisfied at the current state.

 Looking quickly around you, you see a shelf 6, a safe 1, and a drawer
 4. Your task is to: put some vase in safe.
 > go to shelf 6
 On the shelf 6, you see a vase 2.
 > take vase 2 from shelf 6
 You pick up the vase 2 from the shelf 6.
 > go to safe 1
 The safe 1 is closed.
 > open safe 1
 Safe 1 is open. In it, you see a keychain 3.
 > put vase 2 in/on safe 1
 Vase 2 is in/on safe 1.
```

The LLM responds as follows:

```
                             ─── Output ───

 Sub-goals needed to complete the task "put A on B" are

 - Locate object A
 - Find B
 - Ensure B is accessible

 def check_subgoals(state, a, b):
     subgoals_satisfied = 0
     if f"You pick up the {a}" in state:
         subgoals_satisfied += 1
     if f"You arrive at" in state and b in state:
         subgoals_satisfied += 1
     if f"{b} is open" in state:
         subgoals_satisfied += 1
     return subgoals_satisfied
```

Since the LLM is only prompted with one example, it makes a mistake at the last step, which can be fixed by prompting the LLM with "*No need to check if b is open since we only need to put a on b.*". Then the LLM generates a correct rule-based value estimator as follows:

```
                    ┌─ Correct rule-based value estimator ─┐
def check_subgoals(state, a, b):
    subgoals_satisfied = 0
    if f"You pick up the {a}" in state:
        subgoals_satisfied += 1
    if f"You arrive at" in state and b in state:
        subgoals_satisfied += 1
    if f"{a} is in/on {b}" in state:
        subgoals_satisfied += 1
    return subgoals_satisfied
```

