# OpenReview forum: "How Can LLM Guide RL? A Value-Based Approach"
_ICLR.cc/2025/Conference — ICLR 2025 Conference Withdrawn Submission_

### Official Review · Reviewer_E3YS · 2024-10-16

**Soundness:** 2
**Presentation:** 4
**Contribution:** 2
**Rating:** 3
**Confidence:** 4

**Summary:**

In this paper the authors propose to integrate LLMs within the RL framework by introducing a regularization technique that is based on the KL-divergence between the RL agent's policy and the LLM's policy. This method is coined as Language-INtegrated Value Iteration (LINVIT), and the authors introduce a simplified version of it called SLINVIT for empirical evaluation. Under the assumption that the LLM policy is near-optimal, the authors show that this regularization can yield substantial sample efficiency gains for RL agents in several text-based benchmarks.

**Strengths:**

- The presentation of the paper is excellent, with informative figures and algorithms.
- The concept of using LLMs indirectly for regularization is a novel and interesting idea.
- The authors provide a theoretical framework to motivate their decision to introduce this technique, which could serve as a foundation for future work of incorporating LLM priors into RL.
- The authors conducted various experiments to evaluate their proposed methodology.

**Weaknesses:**

- One major concern is that the applicability of the approach is heavily reliant on the LLM policy being essentially (close to) equivalent to the optimal policy. This raises several other concerns.
   - First, the authors argue in line 125 "Moreover, we can still identify the optimal policy for the original MDP even if the LLM's policy is suboptimal", but there are no experiments in the paper showing what happens when the LLM policy is suboptimal.
  -  Nor is any experiment conducted that shows how a baseline LLM policy would perform without the RL agent, nor of the RL agent without LLM regularization.
  - It is likely that if the LLM policy is suboptimal, the performance of the RL agent will be hindered rather than enhanced, which limits the applicability of this approach to environments where the LLM policy is (near) optimal.
  - The authors argue that you need the RL component to be able to adapt and deal with environment feedback, but it seems like the method is only desirable in cases where you have an LLM policy that is almost optimal (which seems like a relatively vague assumption that is not quantifiable in most cases).
- Another concern is that although the paper admits that SLINVIT simplifies the algorithm described in the theoretical section, these simplifications—such as using BFS and single-rollout Monte-Carlo estimation—aren’t well-justified. There’s a disconnect between the theoretical guarantees provided in the KL divergence analysis and the seemingly substantial simplifications made in the experiments where the algorithm is actually implemented.
- In section 5.2 one of the simplifications introduces a process that makes the LLM write python code for functions that yield some value corresponding to a state in the environment. This python code is then subject to a 'one-time human review', which seems like a major impracticality of using this algorithm. Could this review not be done with an LLM as well perhaps to achieve automation?
- The motivations in the experimental section are also rather unclear
  - A brief explanation of the baselines used (e.g., whether they rely on human interventions, purely LLM-driven policies, or RL-based techniques) and how they differ from SLINVIT would help in assessing the fairness of the experiments, as it is currently unclear.
   - The authors introduce an ablation study for the number of Monte-Carlo samples, showing that $M=2$ performs better than $M=1$, yet they proceed to use $M=1$ in the experiments without a clear justification.

**Questions:**

While the paper introduces an interesting concept, it requires significant improvements, particularly in experimental design, explanation of baselines, and addressing concerns about practicality before it can be considered ready for publication.

It is currently unclear if a fair comparison was made to baselines, how it would perform without RL, how it would perform with a sub-optimal LLM policy, how domain-specific it is, how practical the code production and human review is relative to other methods, and whether it could be applied in other environments.

Further minor comments:
- The title should probably be "How can LLMs guide..." instead of "How can LLM guide"
- The paper mentions several times that (e.g. line 49) that "RL agents require huge amounts of random interactions", which comes across as the authors assume that any RL agent uses random action selection for exploration, even though there are much more sophisticated exploration techniques in the literature that do not explore just randomly.
- In line 99 you have the term $V^t_{h+1}(s')$ where $t$ was never defined in the text so I think this is an error you'd like to be aware of.
- In line 108 "the LLM policy is hard to be optimal" should be rephrased.
- Line 110 "priorin" spelling error
- Algorithm 1: I would suggest including where the LLM contributes to the algorithm as it is your main contribution, even if it is with a comment in line 3 of the algorithm.
- It would be nice to have some more details on how exactly you are learning a transition model, this was not very clear to me from the paper.
- In order to measure how optimal the LLM policy is in the given environment, it may be helpful to execute a pure LLM policy for one episode. Based on the accumulated reward, a weighting of the regularization term could be determined.

---

### Official Review · Reviewer_RYPU · 2024-10-19

**Soundness:** 3
**Presentation:** 3
**Contribution:** 3
**Rating:** 6
**Confidence:** 3

**Summary:**

The study introduces
- an information-theoretic alggorithm, LINVIT, which uses LLMs as a regularization factor to enhance the sample efficiency of RL
- a practical version called SLINVIT, simplifying value function construction and using sub-goals to minimize search complexity

Instead of directly using LLMs for decision-making, LINVIT leverages them to refine policy estimation, particularly improving performance when the LLM-provided policy is close to the optimal policy. This approach reduces the data required for learning while retaining the ability to reach the optimal policy even when the LLM policy is suboptimal. Empirical validation in benchmarks like ALFWorld, InterCode, and BlocksWorld shows that LINVIT and SLINVIT significantly outperform traditional RL and LLM methods in terms of sample efficiency and success rates.

**Strengths:**

- This paper provide a new perspective of using LLM to improve the sample efficiency of standard RL methods, which does no require the LLM policy to be optimal. The information-theoretical algorithm, LINVIT, provide a provable guarantee which is favorable.
- Due to hardness in optimization objective in LINVIT, this paper also provide a practical version by replacing the entropy regularizer by directly adding the LLM policy, which provides a practically efficient the algorithm
- Both the theoretical and practical result is quite satisfying, showing that the proposed method is effective.

**Weaknesses:**

- The experiment focuses on the text-based decision problem which LLM is good at, but its performance on more general interative decision-making with abstract state and action is unclear, e.g., solve specific bandit/MDP
- The theory seems standard in analysis of entropy-regularized RL algorithms

**Questions:**

Following the weaknesses,
- Can authors provide more practival motivations? Besides, it seems that the provided algorithm highly relies on value-based RL algorithms, does such insight also make sense when leaverage policy-based methods such as PPO?
- What the techinical contribution/hardness in theoretical analysis?

Also, here's some additional question
- At the end of theoretical analysis, it is claimed that ``the core objective of our theoretical analysis is to justify the use of log-probability as regularization", but SLINVIT uses $\pi_{\rm LLM}$ rather thatn $\log\pi_{\rm LLM}$. Why there's a mismatch? Also, could author provide more elaborations on the connection between KL regularization and practical sum of log probabilities?
- (Minor) What's the difference between notations $\mathbb{P}_{\rm LLM}$ and $\pi_{\rm LLM}$, are these the same?
- (Minor) Is the $a$ missing in $\bar{Q}_h^t(s,a)$ for (3.4)?

---

### Official Review · Reviewer_VPm2 · 2024-10-24

**Soundness:** 2
**Presentation:** 1
**Contribution:** 1
**Rating:** 3
**Confidence:** 3

**Summary:**

The paper aims improve sample efficiency of reinforcement learning by incorporating prior knowledge from LLMs with a novel regularization technique for the policy and value function inspired by entropy-regularized MDPs.

**Strengths:**

- The empirical results shows that the proposed methodology outperforms the chosen baselines.
- The proposed methodology is evaluated on multiple benchmarks.

**Weaknesses:**

1. The structure of the paper makes it hard to follow since it jumps from the theoretical justification of the method to the presentation of the results to then go back to a somewhat related theoretical result. A better division of the theory and practice would make the paper easier to follow.
2. The proposed methodology is particularly weak as one version (**rule-based**) strictly requires a human review each time the agent is run and the other (**Monte Carlo**) is not effectively evaluated as the number of sample is set to 1 in the experiments.
3. Theorem 6.2 is a generic proof that is not particularly relevant to the proposed methodology but rather to approximation errors.

**Questions:**

1. Where is Eq. 3.1 taken from? It is presented as a specific choice of the author but it is surely defined in a different theoretical context.
2. I assume that the LINVIT is the theoretical motivation of SLINVIT, however it cannot act as so if it's not empirically validated. Have you tried running LINVIT in a tabular reinforcement learning setting? How can we know if the methodology is sound?
3. The rule-based method requires a human review, how is this process handled? Do you have guidelines on how this should be done? What if the LLMs proposed a value function that looks wrong but it correctly compiles?

**Details Of Ethics Concerns:**

No ethical concerns for this submission.

---

### Official Review · Reviewer_idD7 · 2024-11-04

**Soundness:** 2
**Presentation:** 2
**Contribution:** 2
**Rating:** 5
**Confidence:** 4

**Summary:**

Large language models (LLMs) have demonstrated remarkable proficiency in language understanding and generation. However, they are limited in exploration and self-improvement for planning tasks, as they lack autonomous refinement based on feedback. This paper explores how leveraging the policy prior from LLMs can enhance the sample efficiency of reinforcement learning (RL) algorithms.

**Strengths:**

1. This paper introduces a novel approach to improve sample efficiency in RL by integrating LLMs, avoiding the need for fine-tuning the LLM and demonstrating superior performance compared to prompt-engineering methods, such as reflextion and ReAct.
2. It provides structured sub-goals for tasks in ALFWorld, with fine-grained value estimation that supports effective task-solving.
3. The paper includes a theoretical analysis of the relationship between KL divergence and policy performance, offering insights that an LLM policy closer to the optimal policy can help solve RL tasks more efficiently.

**Weaknesses:**

1. The implementation details of value-based methods are unclear. Section 5.2 mentions rule-based and Monte Carlo (MC) value estimates, but neither aligns directly with traditional value-based RL algorithms. Additionally, it is unclear which methods were applied to specific benchmarks, and the study lacks ablation analysis for different value estimation methods.
2. Although the paper provides a theoretical analysis of the connection between KL divergence and the policy performance bound, it would benefit from an empirical demonstration of this relationship, even on a simple tabular task. To achieve this, the KL divergence between the LLM and the optimal policy, as well as the value difference between the LLM-guided policy and the optimal policy, should be provided.
3. The motivation for aligning the learned policy with the LLM (e.g., maximizing -KL as in Eq. 3.2) is unclear. In traditional RL (e.g., TRPO), a KL constraint is used to ensure robust learning, so further clarification on why adding a kl divergence is desirable would strengthen this paper.
4. The explanation of simplifying KL divergence as the probability of \(\pi^{LLM}\) is insufficient. It would be helpful to discuss why this simplification maintains equivalence to the original KL and to explore its potential impact.

**Questions:**

Refer to the weaknesses mentioned above.

Additionally, the RAP framework utilizes MCTS for Blocksworld tasks, also use MC sampling and "likelihood of action" as a reward, similar to the approach in Eq. 5.1 of this paper. Given these similarities, what is the primary reason for the substantial performance advantage of SLINVIT over RAP?

---

### Note · Authors · 2024-11-27

I have read and agree with the venue's withdrawal policy on behalf of myself and my co-authors.